# Metformin, Statin Use, and Female Colorectal Cancer: A Population-Based Cohort Study in Taiwan

**DOI:** 10.3390/jcm11154469

**Published:** 2022-07-31

**Authors:** Shu-Hua Hsu, De-Kai Syu, Yu-Chiao Wang, Chih-Kuang Liu, Ming-Chih Chen

**Affiliations:** 1Department of Family Medicine, Fu Jen Catholic University Hospital, Fu Jen Catholic University, New Taipei City 243, Taiwan; crazysunnyhsu@gmail.com; 2Department of Orthopedics, Fu Jen Catholic University Hospital, Fu Jen Catholic University, New Taipei City 243, Taiwan; gladiator711124@gmail.com; 3Graduate Institute of Business Administration, College of Management, Fu Jen Catholic University, New Taipei City 242, Taiwan; 406088139@mail.fju.edu.tw; 4Department of Urology, Fu Jen Catholic University Hospital, Fu Jen Catholic University, New Taipei City 243, Taiwan; 5Artificial Intelligence Development Center, Fu Jen Catholic University, New Taipei City 242, Taiwan

**Keywords:** colorectal cancer, population-based cohort study, metformin, statin

## Abstract

In the last few years, the incidence of colorectal cancer (CRC) in women has gradually increased. However, epidemiological studies on the relationship between type II diabetes mellitus (T2DM) and female CRC and the effect of metformin or statins on female CRC are insufficient. To determine their association, we conducted a population-based cohort study on women in Taiwan. We collected data on a total of 396,521 women aged 40 to 64 years old from 1 January 2007 to 31 December 2009 from the National Health Insurance Research Database. We followed up on all participants in the cohort until the occurrence of CRC, the date for all death, or 31 December 2015. Full development of CRC was identified using the International Classification of Disease (ICD), 9th Revision, code 153. We estimated hazard ratios (HRs) and corresponding 95% confidence intervals (CIs) using the Cox proportional hazards model. Both metformin (adjusted hazard ratio, aHR = 1.12; 95% CI: 0.934–1.335, *p* = 0.227) and statin (aHR = 1.03; 95% CI: 0.906–1.172, *p* = 0.645) use showed no association with female CRC in a multivariate analysis. The findings indicate that metformin and statin use showed no protective effect against female colorectal cancer (CRC). An additional randomized trial is necessary to investigate the effect of metformin and statin use in CRC prevention.

## 1. Introduction

Colorectal cancer (CRC) is the third most common malignancy and the second most common cause of cancer death worldwide [1]. The epidemiological characteristics and risk factors of major types of cancers vary in each country [2]. In Taiwan, malignant neoplasms have been the leading cause of death since 1982 [3]. As the Taiwanese lifestyle has become more westernized, the mortality rate from diseases such as CRC has gradually increased for males and females since 1971 [2]. CRC is primarily considered a “lifestyle” disease. Demographic factors including age, gender, genetics, the consumption of high-calorie foods and diets high in animal fat, alcohol consumption, and obesity are considered to be potential risk factors [4]. Vulnerability differences in CRC may be divergent due to gender and sex hormones [5]. CRC is the second most common cancer in females in Taiwan [6]. For women, the lifetime risk of developing colorectal cancer is 6% and the five-year survival rate of colorectal cancer among women is lower than that among men [7]. The above epidemiological evidence implies that colorectal cancer is a major health threat for women, although it has often been underestimated.

Type 2 diabetes mellitus (T2DM) is a risk factor for solid malignancies such as liver, pancreatic, colon, breast, bladder, and endometrial cancer [8,9,10,11]. The cancer-causing mechanisms in diabetes are complex, including excessive reactive oxygen species (ROS) formation, chronic inflammation, and impaired healing phenomena, collectively leading to carcinogenesis under diabetic conditions [12,13]. Metformin is a first-line anti-diabetic drug, and study results on the relationship between metformin use and CRC are controversial and conflicting [14]. A case-control study showed that there was no association between metformin use and the incidence of CRC in T2DM patients (hazard ratio (HR) 0.96; 95% confidence intervals (CI) 0.88–1.04) [15]. However, a meta-analysis suggested that metformin users had a significantly lower incidence of CRC (relative risk (RR) 0.76, 95% CI: 0.69–0.84, *p* < 0.001) compared to non-metformin users [16].

Epidemiological studies have reported that increased cholesterol levels are associated with higher cancer incidence, including CRC [17,18,19,20]. Statins remain the first-line treatment for managing dyslipidemia as they can inhibit the cell cycle and induce apoptosis; however, there has also been debate as to the effect of statins on CRC. [21]. Lee et al. found that compared to non-users, the adjusted hazard ratio (aHR) for risk of CRC in high statin users was 0.56 (CI: 0.42–0.75) in men and 0.64 (CI: 0.46–0.90) in women [22]. To our knowledge, there are controversial results and a lack of conclusive evidence in previous literature about the influence of metformin and statin on CRC. Therefore, this study aimed to investigate the effect of metformin and statin use on female CRC.

## 2. Materials and Methods

### 2.1. Data Source and Study Population

This was a propensity score-matched, retrospective cohort study based on a population of women aged 40–64 years in Taiwan. The National Health Insurance (NHI) data. base, the Taiwan Cancer Registry (TCR), and the National Cause of Death Registry (NCDR) database, which are all available from the Health and Welfare Data Science Center (HWDC), were used in this study. The NHI database is an electronic, de-identified, administrative healthcare database containing the medical records of all outpatients and inpatients [23]. The national population-based TCR incorporates the diagnosis date of pathologically confirmed cancer cases into the registry and has high-quality data [24]. Following the center’s ethical guidelines, patients’ personal information was anonymized before we accessed it; consequently, the Research Ethics Committee waived the requirement for informed consent. This study was approved by the Institutional Review Board (IRB) and Ethics Committee of Fu Jen Catholic University (IRB approval no.: C107099).

### 2.2. Participant Selection

Figure 1 shows a flowchart of the selection procedure used in this study. A total of 4,018,833 Taiwanese women aged 40 to 64 (as CRC is rare in young people) who sought medical treatment from 2007 to 2009 were initially enrolled in the cohort and were divided into diabetic and non-diabetic groups based on disease code and medication prescription. The diabetic group was defined by records of at least three outpatient or inpatient diabetes diagnoses and prescribed antidiabetic medications within one year. The definition was according to the previous study from the Taiwan NHI Research Database [25], and the first diagnosis date was regarded as the index date. The other patients were classified into the non-diabetic group, and the date of their first visit to the doctor was regarded as the index date. Patients who had had type 1 DM and CRC prior to the index date were excluded from our study. Definitions related to disease diagnosis and drugs are listed in Appendix A Table A1, Table A2 and Table A3 based on a previously published study in Taiwan [26].

### 2.3. Propensity Score Matching and Covariates

To reduce the effects of potential confounders and sampling bias in the diabetic and non-diabetic groups, this research used a 1:4 ratio in propensity score matching (PSM). This is a common technique for selecting controls with identical background covariates to minimize differences among groups of study participants. The matching variables used were age, comorbidities (hypertension, chronic obstructive pulmonary disease (COPD), asthma, stroke, nephropathy, ischemic heart disease (IHD), peripheral arterial disease (PAD), eye disease, dyslipidemia, and obesity), concomitant medications (fibrates, angiotensin-converting enzyme inhibitor (ACEI), angiotensin receptor blocker (ARB), calcium channel blockers (CCB), aspirin, dipyridamole, clopidogrel, ticlopidine, and other non-steroidal anti-inflammatory drugs (NSAIDs)), and potential colon cancer detection examinations (definitions are listed in Appendix A Table A1, Table A2 and Table A3 and see Table 1). After PSM, the final cohort of patients with T2DM comprised 86,992 patients, whereas the comparison cohort contained 309,529 patients.

### 2.4. Statistical Analyses

Basic demographic characteristics were defined as categorical (presented as N (%)) and continuous variables (presented as mean ± standard deviation (SD)) in this study. The primary outcome of this study was the end point of CRC (ICD, 9th Revision, code 153). The person-years of follow-up (censoring time) for each participant were calculated from the index date until the occurrence of CRC, the date for all death, or the last date of linked data available from the TCR and NCDR (31 December 2015), whichever occurred first.

We used a Cox proportional hazards regression to calculate hazard ratios (HRs) with 95% confidence intervals (CIs) between each risk factor and CRC. Variables were considered for more than three records in one year during the five years before the index date, including hypertension, chronic obstructive pulmonary disease (COPD), asthma, stroke, nephropathy, ischemic heart disease (IHD), peripheral arterial disease (PAD), eye disease, dyslipidemia, obesity, statin, fibrates, angiotensin-converting enzyme inhibitor (ACEI), angiotensin receptor blocker (ARB), calcium channel blockers (CCB), aspirin, dipyridamole, clopidogrel, ticlopidine, non-steroidal anti-inflammatory drugs (NSAIDs), sulfonylurea, insulin, acarbose, thiazolidinedione (TZD), potential colon cancer detection examinations, and metformin use (see Appendix A Table A1, Table A2 and Table A3).

Even if PSM is applied, residual imbalance might still exist in a population [27], and a multivariate Cox regression analysis should still be performed. Therefore, we performed a multivariate analysis for variables that were statistically significant in univariate analysis. The results of the 95% CI of HR were also presented visually as forest plots. The life table method was used to estimate the cumulative incidence of CRC per year of follow-up among subjects who had used metformin and statins prior to enrollment. All p values < 0.05 were considered statistically significant. The SAS statistical package (version 9.4; SAS Institute, Inc., Cary, NC, USA) and STATA (version 16.0; StataCorp, College Station, TX, USA) were used for all data analyses.

## 3. Results

A total of 396,521 women were included in this study. The number of CRCs developed during the observation period was 665 in the diabetes group and 2192 in the non-diabetes group (Figure 1). Table 1 displays the baseline characteristics of the female patients aged 40 to 64 with and without DM. After propensity score matching with index year, age, multiple chronic diseases, and drugs, there was little difference between the non-DM and DM groups. The prevalence of nephropathy, PAD, and eye disease was higher in the DM group compared to the non-DM group. These factors were comorbidities of DM, which may cause a confounding effect in this study.

Table 2 shows the risk factors associated with CRC incidence. The univariate analysis results showed that age (every one-year increment, crude HR = 1.07; 95% CI: 1.050–1.083), diabetes status (crude HR = 1.22; 95% CI: 1.112–1.334), nephropathy status (crude HR = 1.29; 95% CI: 1.033–1.621), statin use (crude HR = 1.12; 95% CI: 1.015–1.245), calcium channel blocker use (crude HR = 1.18; 95% CI: 1.000–1.401), other NSAIDs use (crude HR = 0.71; 95% CI: 0.565–0.894), sulfonylurea use (crude HR = 1.23; 95% CI: 1.121–1.345), and metformin use (crude HR = 1.26; 95% CI: 1.147–1.379) were statistically significantly associated with CRC.

Figure 2 shows the cumulative incidence of female colorectal cancer based on metformin and statin use. For the non-metformin and non-statin use groups, the median follow-up period was 8.76 years. During the observation period, the Nelson–Aalen estimate of the cumulative incidence of female CRC was 0.0004% in one year, 0.0015% in three years, 0.0029% in five years, and 0.0056% in eight years. For both the metformin and statin use groups, the median follow-up period was 8.18 years. The cumulative incidence of female CRC was 0.0006% in one year, 0.0023% in three years, 0.0047% in five years, and 0.0088% in eight years. The cumulative incidence in the metformin and statin use groups was significantly higher than that in the non-metformin and non-statin use groups (log-rank test *p* < 0.001; Figure 2).

Table 3 shows the combined effects of statins and metformin. When adjusted for multiple comparisons, age (aHR = 1.07; 95% CI: 1.052–1.085, *p* < 0.001), nephropathy status (aHR = 1.31; 95% CI: 1.040–1.641, *p* = 0.022), CCB use (aHR = 1.19; 95% CI: 1.007–1.416, *p* = 0.042), and other NSAIDs use (aHR = 0.73; 95% CI: 0.575–0.915, *p* = 0.007) were statistically significantly associated with CRC. Additionally, we assessed the combination effect of metformin and statin use on CRC incidence. Compared to the non-metformin use and non-statin use groups, the risk of CRC was positively associated with the metformin and statin use groups (aHR = 1.25; 95% CI: 1.043–1.499, *p* = 0.016).

## 4. Discussion

This is a nationwide retrospective cohort study. The accuracy of the NHI database guaranteed appropriate statistical power for detecting the association between two diseases and completeness in identifying incident cases of female CRC. We determined the characteristics of CRC among the Taiwanese female population and the fact that metformin and statin use showed no protective effect on the risk of CRC in women. 

In our study, the risk of colon cancer in patients with T2DM remained higher than that in patients without T2DM in the Cox regression analyses. This result is consistent with the results of previous studies, which showed that T2DM was a precipitating factor of CRC. Soltani et al. demonstrated a significant association between suffering from T2DM and having colon adenoma, which is a pre-cancerous condition [28]. A prospective study of two US cohorts showed that T2DM was statistically significantly associated with an increased risk of CRC [29]. A meta-analysis analyzing 151 cohort studies comprising 32 million people showed that T2DM was associated with an increased risk of CRC [8].

In the multivariate analysis, diabetes status showed no statistically significant difference with regard to CRC (Table 3). The incidence of nephropathy, PAD, and eye diseases was higher in the DM group, and these variables were cofactors of DM. Diabetes’ effect on CRC may be diluted in multivariate analysis. Comorbidity, such as nephropathy, was also associated with a higher risk of CRC both in univariate and multivariate analysis (Table 2 and Table 3). Kidney function is damaged by poor long-term glycemic control, and so diabetic nephropathy is a comorbidity of T2DM. It is currently the leading cause of chronic kidney disease and end-stage renal disease [30]. The relationship between PAD and DM neuropathy also implied a complex link between diabetes and CRC, which may be influenced by the severity of diabetes. Thus, we included these chronic comorbidities in this study even if they may cause confounding effects. The result would have been more conservative if we had excluded these variables.

The effect of metformin on CRC is inconsistent between different studies. Knapen et al. revealed that metformin users had a 1.2-fold increased risk of colorectal cancer compared with a non-user [31]. Bodmer et al. showed that metformin was associated with a slightly increased risk of colorectal cancer compared with non-users [32]. Some studies have reported that there is no association between metformin use and CRC [33,34]. However, other studies have shown a protective effect. In a meta-analysis report, which includes eight cohort studies and three case-control studies, metformin was associated with a 25% reduction in CRC incidence among T2DM patients [35]. Another meta-analysis showed that metformin intake was associated with a 25% reduction in colorectal adenoma incidence and a 22% decrease in CRC risk in T2DM metformin users when compared with T2DM non-metformin patients [36].

In our study, metformin showed no protective effect on female CRC. From a pathophysiological perspective, since the primary actions of metformin significantly reduce circulating glucose and plasma insulin, hence improving insulin resistance, it may be beneficial for reducing the risk of diabetes-related cancer incidents [37]. However, elevated glucose levels lead to the proliferation of various solid tumor cell lines and play a role in the development of cancer [8,38]. Some studies have revealed that there is a positive association between serum glucose and the risk of cancer. Stocks et al. examined six European cohorts, analyzing 274,126 men and 275,818 women, which showed that for one mmol/l increase in blood glucose, the relative risk of cancer in men was 1.05 and in women was 1.11 [39]. A study in Austria, which included more than 140,000 adults with an average of 8.4 years’ follow-up, showed that high fasting blood glucose was associated with several cancers, including CRC in women [40]. In addition, a meta-analysis demonstrated that high levels of glucose and hemoglobin A1C (HbA1c) were related to incidences of CRC [41].

NSAIDs use showed a protective effect on CRC in our study due to its anti-inflammatory effect [42]. We found that dyslipidemia was not statistically significantly correlated with female CRC, and the age range was 40 to 64 years. According to previous literature, the risk of CRC is also related to estrogen exposure and the severity of dyslipidemia. The effect of statins on CRC has also shown conflicting results in previous studies. There is an increasing body of data supporting an inverse association between the use of statins and the mortality rate in CRC [43]. However, a meta-analysis revealed five publications on statins and summary RRs which reported that there was no association between statin use and CRC incidence compared to non-use [44]. Yanqiong Liu et al. suggest that statin use is associated with a modestly reduced risk of CRC. However, long-term statin use did not appear to significantly affect the risk of CRC [45]. In our study, statins had a slightly positive association with CRC in the univariate analysis and showed no association with CRC in the multivariate analysis. The potential protective effect of statins may also be weakened by poor T2DM control.

Our study has several limitations. This cohort included only Taiwanese women; some studies showed that there may be ethnic differences in the etiology and biology of CRC between Asians and non-Asians. Therefore, the generalization of our study findings to other ethnicities should be reconfirmed. We did not have biochemical data such as glucose, HbA1c, or lipid profiles to evaluate their potential effects; however, DM severity, dyslipidemia severity, or glycemic control status is related to the risk of CRC and can be further analyzed. Moreover, the duration of insulin use was related to T2DM severity, and it would be worth exploring the effect of insulin on CRC in future studies. It is difficult to evaluate newly detected T2DM and dyslipidemia, as their status could change during the follow-up period. Therefore, we might have underestimated the relative risk of CRC associated with T2DM and dyslipidemia. Some potential confounders such as dietary factors, physical activity, and family history were not measured. As this is an observational study, an experimental design is necessary to confirm the effect of metformin and statin use on the development of CRC.

## 5. Conclusions

This study found that metformin and statin use revealed no protective effect on female CRC. Poor glycemic control may increase the risk of CRC and may weaken the possible effects of metformin or statin use on CRC, and an additional randomized trial is necessary to investigate the effect of metformin and statin use on CRC prevention. It is important to prevent metabolic syndrome and aggressively control blood glucose and HbA1C to fall within adequate levels in T2DM patients for the first-degree prevention of CRC. Our study supports public health initiatives to combat the increased prevalence of T2DM and CRC among the Taiwanese population.

## Figures and Tables

**Figure 1 jcm-11-04469-f001:**
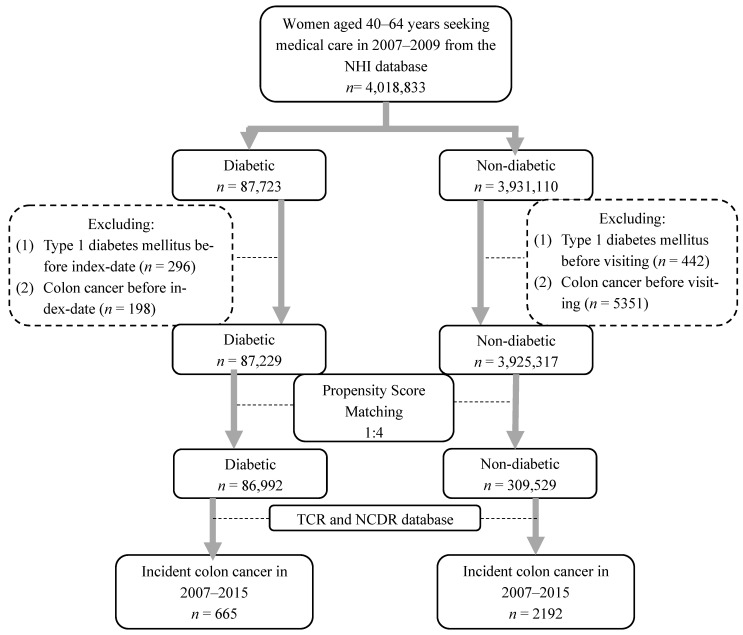
Flow chart for the selection of study subjects.

**Figure 2 jcm-11-04469-f002:**
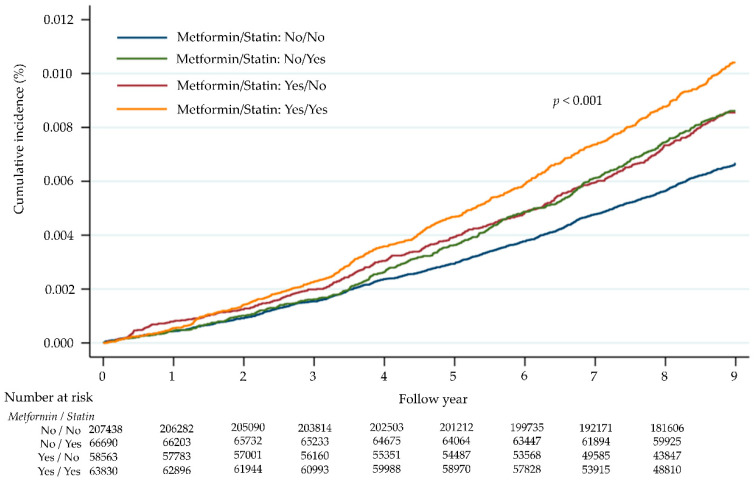
Cumulative incidence of female colorectal cancer based on different scenarios of metformin and statin use.

**Table 1 jcm-11-04469-t001:** Baseline characteristics of female patients aged 40–64 years with and without diabetes mellitus.

Variables	Unmatched			Matched to Propensity Scores		
	non-DM (*n* = 3,925,317)	DM (*n* = 87,229)	Standardized Difference	non-DM (*n* = 348,916)	DM (*n* = 87,229)	Standardized Difference
Age	48.96 ± 7.01	54.36 ± 6.2	0.816	54.47 ± 6.36	54.35 ± 6.20	−0.018
Hypertension	581,079 (14.8%)	51,811 (59.4%)	−1.041	185,152 (59.8%)	51,585 (59%)	0.012
COPD	215,797 (5.5%)	10,439 (12%)	−0.231	38,735 (12.5%)	10,411 (12%)	0.019
Asthma	113,219 (2.9%)	5041 (5.8%)	−0.143	19,331 (6.2%)	5021 (5.8%)	0.023
Stroke	73,235 (1.9%)	8647 (9.9%)	−0.347	28,734 (9.3%)	8617 (9.9%)	−0.027
Nephropathy	46,121 (1.2%)	6302 (7.2%)	−0.305	16,702 (5.4%)	6268 (7.2%)	−0.091
IHD	163,156 (4.2%)	15,857 (18.2%)	−0.457	56,336 (18.2%)	15,803 (18%)	0.001
PAD	46,858 (1.2%)	10,344 (11.9%)	−0.442	26,764 (8.6%)	10,281 (12%)	−0.132
Eye Disease	13,145 (0.3%)	13,940 (16%)	−0.596	12,961 (4.2%)	13,707 (16%)	−0.441
Dyslipidemia	305,862 (7.8%)	44,911 (51.5%)	−1.090	157,819 (51%)	44,694 (51%)	−0.010
Obesity	16,269 (0.4%)	1829 (2.1%)	−0.152	6553 (2.1%)	1824 (2.1%)	0.002
Fibrates	76,635 (2%)	17,323 (19.9%)	−0.600	52,656 (17%)	17,271 (20%)	−0.095
ACEI/ARB	310,000 (7.9%)	43,437 (49.8%)	−1.043	143,766 (46.4%)	43,212 (50%)	−0.080
CCB	403,407 (10.3%)	37,991 (43.6%)	−0.809	132,139 (42.7%)	37,839 (44%)	−0.020
Aspirin	299,482 (7.6%)	31,040 (35.6%)	−0.722	104,535 (33.8%)	30,960 (36%)	−0.047
Dipyridamole	95,552 (2.4%)	12,136 (13.9%)	−0.428	37,880 (12.2%)	12,104 (14%)	−0.063
Clopidogrel/Ticlopidine	15,380 (0.4%)	3274 (3.8%)	−0.238	8422 (2.7%)	3251 (3.7%)	−0.072
Other NSAIDs	2,797,482 (71.3%)	73,193 (83.9%)	−0.307	260,994 (84.3%)	72,976 (84%)	0.010
Potential colon cancer detection examinations	2179 (0.1%)	88 (0.1%)	−0.016	306 (0.1%)	88 (0.1%)	−0.001

Abbreviations: COPD, chronic obstructive pulmonary disease; IHD, ischemic heart disease; PAD, peripheral arterial disease; ACEI, angiotensin-converting enzyme inhibitor; ARB, angiotensin receptor blocker; CCB, calcium channel blocker; NSAIDs, non-steroidal anti-inflammatory drugs.

**Table 2 jcm-11-04469-t002:** Risk factors associated with colon cancer incidence for women aged 40 to 64 years.

Variables	Interpretation	Colon Cancer Cases (*n*)	Univariate		
			HR ^a^	95% CI	*p* Value
Age	Every 1-year increment	2857	1.07	(1.050, 1.083)	<0.001
Diabetes	Yes vs. no	665/2192	1.22	(1.112, 1.334)	<0.001
Hypertension	Yes vs. no	1930/927	1.18	(0.951, 1.474)	0.131
COPD	Yes vs. no	374/2483	0.93	(0.774, 1.122)	0.456
Asthma	Yes vs. no	206/2651	1.02	(0.809, 1.295)	0.845
Stroke	Yes vs. no	290/2567	1.04	(0.847, 1.277)	0.708
Nephropathy	Yes vs. no	227/2630	1.29	(1.033, 1.621)	0.025
IHD	Yes vs. no	607/2250	1.11	(0.942, 1.305)	0.213
PAD	Yes vs. no	265/2592	0.82	(0.667, 1.014)	0.068
Eye Disease	Yes vs. no	237/2620	1.13	(0.852, 1.498)	0.396
Dyslipidemia	Yes vs. no	1608/1249	0.96	(0.780, 1.184)	0.709
Obesity	Yes vs. no	58/2799	1.05	(0.722, 1.528)	0.798
Statin	Yes vs. no	1127/1730	1.12	(1.015, 1.245)	0.024
Fibrates	Yes vs. no	554/2303	1.13	(0.953, 1.327)	0.165
ACEI/ARB	Yes vs. no	1544/1313	1.15	(0.936, 1.403)	0.186
CCB	Yes vs. no	1470/1387	1.18	(1.000, 1.401)	0.0498
Aspirin	Yes vs. no	1068/1789	0.91	(0.775, 1.070)	0.254
Dipyridamole	Yes vs. no	434/2423	1.02	(0.849, 1.228)	0.824
Clopidogrel/Ticlopidine	Yes vs. no	115/2742	1.32	(0.995, 1.746)	0.054
Other NSAIDs	Yes vs. no	2363/494	0.71	(0.565, 0.894)	0.004
Sulfonylurea	Yes vs. no	1054/1803	1.23	(1.121, 1.345)	<0.001
Metformin	Yes vs. no	1027/1830	1.26	(1.147, 1.379)	<0.001
Insulin	Yes vs. no	192/2665	1.14	(0.941, 1.373)	0.183
Acarbose	Yes vs. no	282/2575	1.12	(0.966, 1.309)	0.130
TZD	Yes vs. no	333/2524	1.02	(0.882, 1.175)	0.809
Potential colon cancer detection examinations	Yes vs. no	2/2857	0.68	(0.124, 3.734)	0.657

Abbreviations: COPD, chronic obstructive pulmonary disease; IHD, ischemic heart disease; PAD, peripheral arterial disease; ACEI, angiotensin-converting enzyme inhibitor; ARB, angiotensin receptor blocker; CCB, calcium channel blocker; NSAIDs, non-steroidal anti-inflammatory drugs; TZD, thiazolidinedione; HR, hazard ratio; CI, confidence interval. ^a^: Adjusted for age via a Cox proportional hazards regression.

**Table 3 jcm-11-04469-t003:** Multivariate analysis of the incidence of colon cancer in women aged 40–64 years with the combination of statins and metformin.

Variables	Items	HR ^a^ (95% CI)		95% CI of HR	*p* Value
Age		1.07	(1.052, 1.085)	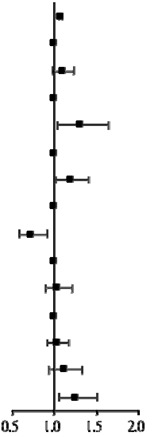	<0.001
Diabetes	No	1.00	Referent		
	Yes	1.09	(0.977, 1.223)		0.119
Nephropathy	No	1.00	Referent		
	Yes	1.31	(1.040, 1.641)		0.022
CCB	No	1.00	Referent		
	Yes	1.19	(1.007, 1.416)		0.042
Other NSAIDs	No	1.00	Referent		
	Yes	0.73	(0.575, 0.915)		0.007
Sulfonylurea	No	1.00	Referent		
	Yes	1.04	(0.892, 1.214)		0.614
Metformin/ Statin	No/No	1.00	Referent		
	No/Yes	1.03	(0.906, 1.172)		0.645
	Yes/No	1.12	(0.934, 1.335)		0.227
	Yes/Yes	1.25	(1.043, 1.499)		0.016

^a^ Multiple Cox regression. Covariates: age, diabetes, nephropathy, CCB, other NSAIDs, sulfonylurea, metformin, and statin.

## Data Availability

The datasets generated and/or analyzed in this study are not publicly available in accordance with the policy of the Health and Welfare Data Science Center, Ministry of Health and Welfare, Taiwan, but are available from the corresponding author upon reasonable request.

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
