# Peer review of "Metformin, Statin Use, and Female Colorectal Cancer: A Population-Based Cohort Study in Taiwan"

_jcm, 2022, doi:10.3390/jcm11154469_

Round 1

Reviewer 1 Report

Hsu et al. investigated the incidence of colorectal cancer in females, and compared the differences in those having T2DM vs non-T2DM patients. Although the study is of interest, both the manuscript and study design has serious issues that must be revised.

Major issues:

1. Besides the brief description of the study population, the manuscript lacks the description of the parameters investigated, statistical methods, etc. 

2. Due to the restrictions authors defined T2DM patients, the number of T2DM patients included in the study is very limited. In most areas of the world, between the age of 40-64, the percentage of T2DM patients is higher than the ~2% of the current study. Definition of T2DM must be redefined.

3. What was the rationale behind using propensity score matching? If authors want to conduct a nationwide analysis, the statistical analysis must be done on the whole population, not in a specific subset.

4. Due to the restrictions used to define T2DM patients, the ones, who did not meet these definitions can cause bias if they were included in the non-T2DM population. As authors used propensity score matching, and these "non-T2DM" patients are probably much better matches compared to those of true non-T2DM ones, the results may have been introduced with a significant bias.

5. Baseline description of the study population is missing. Also, the number of CRCs develeoped in the observation period is missing as well. From Table 2, one could assume this number is 2857, however, considering the long follow-up time and large baseline population, this number seem to bee too small (similar to that of the number of T2DM patients).

6. In Discussion, authors wrote that metformin usage was compared to non-diabetcs. Please explain, why? Metformin usage should be compared within the the diabetic subpopulation only!

7. Authors should consider dividing NSAID to ASA (aspirin) and other NSAIDs.

8. Based on T2DM medication, the advancement of T2DM could be predicted. It would be interesting to compare the different groups of T2DM patients in relation to CRC risk.

9. If authors can obtain the diagnosis date for T2DM from the database, the relationship between CRC and T2DM duration would be an interesting question.

10. Figure 2: If the curves "touch" or cross each other, that is not cumulative incidence!

Minor issues:

1. There is a newer GLOBOCAN report.

2. There are some abbreviatoins throughout the manuscript, which were never solved within the main text (e.g., aHR and PAD).

3. Authors wrote that the result on metformin vs CRC is controversial. This should be clarified as it is not limited to CRC only, but to cancer in general.

4. In discussion it is not recommended to repeat numeric results and p-values.

5. An English proof-reading of a native speaker would be beneficial.

Author Response

Major issues:

Point 1: Besides the brief description of the study population, the manuscript lacks the description of the parameters investigated, statistical methods, etc.

Response 1: Dear reviewer, thanks for your comment and advice. Based on the reviewer's comprehensive suggestion, in Page 2-4, Line 73-145, we revised the methodology section and made more detailed additions:

  1. Materials and Methods

2.1. Data Source and Study Population

This was a propensity score-matched, retrospective cohort study based on a population of. women aged 40–64 years in Taiwan. The National Health Insurance (NHI) data. base, the Taiwan Cancer Registry (TCR), and the National Cause of Death Registry (NCDR) database, which are all available from the Health and Welfare Data Science Center (HWDC), were used in this study. The NHI database is an electronic, de-identified, administrative healthcare database containing the medical records of all outpatients and inpatients[23]. The national population-based TCR incorporates the diagnosis date of pathologically confirmed cancer cases into the registry and has high quality data [24]. Following the center's ethical guidelines, patients’ personal information was anonymized before we accessed it; consequently, the Research Ethics Committee waived the requirement for informed consent. This study was approved by the Institutional Review Board (IRB) and Ethics Committee of Fu Jen Catholic University (IRB approval no.: C107099).                       

2.2. Participant Selection

Figure 1 shows a flowchart of the selection procedure used in this study. A total of 4,018,833 Taiwanese women aged 40 to 64 (as CRC is rare in young people) who sought medical treatment from 2007 to 2009 were initially enrolled in the cohort, and was divided into diabetic and non-diabetic groups based on disease code and medication prescription. The diabetic group was defined by records of at least three outpatient or inpatient diabetes diagnosis and prescribed antidiabetic medications within one year. The definition was according to the previous study from the Taiwan NHI Research Database[25], and the first diagnosis date was regarded as the index date. The other patients were classified into the non-diabetic group, and the date of their first visit to the doctor was regarded as the index date. Patients who had had type 1 DM and CRC prior to the index date were excluded from our study. Definitions related to disease diagnosis and drugs are listed in Appendix Documents Tables A–C based on a previously published study in Taiwan[26].

2.3. Propensity Score Matching and Covariates

To reduce the effects of potential confounders and sampling bias in the diabetic and non- diabetic groups, this research used a 1:4 ratio in propensity score matching(PSM) . It is a common technique for selecting controls with identical background covariates to minimize differences among groups of study participants. The matching variables used were age, comorbidities (hypertension, chronic obstructive pulmonary disease (COPD), asthma, stroke, nephropathy, ischemic heart disease (IHD), peripheral arterial disease (PAD), eye disease, dyslipidemia, and obesity), concomitant medications (fibrates, angiotensin-converting enzyme inhibitor (ACEI), angiotensin receptor blocker (ARB), calcium channel blockers (CCB), aspirin, dipyridamole, clopidogrel, ticlopidine, and other non-steroidal anti-inflammatory drugs (NSAID)), and potential colon cancer detection examinations (definitions are listed in Appendix Documents Tables A–C and see Table 1). After PSM, the final cohort of patients with T2DM comprised 86,992 patients, whereas the comparison cohort contained 309,529 patients.

2.4. Statistical analyses

Basic demographic characteristics were defined as categorical (presented as N (%)) and con. tinuous variables (presented as mean ± standard deviation (SD)) in this study. The primary outcome in this study was the end point of CRC (ICD, 9th Revision, code 153). The person-years of follow-up (censoring time) for each participant were calculated from the index date until the occurrence of CRC, the date for all death, or the last date of linked data available from the TCR and NCDR (December 31, 2015), whichever occurred first.

We used a Cox proportional hazards regression to calculate hazard ratios (HRs) with 95% confidence intervals (CIs) between each risk factor and CRC. Variables were considered for more than three records in one year during the five years before the index date, including hypertension, chronic obstructive pulmonary disease (COPD), asthma, stroke, nephropathy, ischemic heart disease (IHD), peripheral arterial disease (PAD), eye disease, dyslipidemia, obesity, statin, fibrates, angiotensin-converting enzyme inhibitor (ACEI), angiotensin receptor blocker (ARB), calcium channel blockers (CCB), aspirin, dipyridamole, clopidogrel, ticlopidine, non-steroidal anti-inflammatory drugs (NSAID), sulfonylurea, insulin, acarbose, thiazolidinedione (TZD), potential colon cancer detection examinations, and metformin use. (see Appendix Documents Tables A–C).

Even if PSM is applied, residual imbalance might still exist in a population [27], and a multivariate Cox regression analysis should still be performed. Therefore, we performed a multivariate analysis for variables that were statistically significant in univariate analysis. The results of the 95% CI of HR were also presented visually as forest plots. The life table method was used to estimate the cumulative incidence of CRC per year of follow-up among subjects who had used metformin and statins prior to enrollment. All p values < 0.05 were considered statistically significant. The SAS statistical package (version 9.4; SAS Institute, Inc., Cary, NC, USA) and STATA (version 16.0; StataCorp, College Station, TX, USA) were used for all data analyses.analysis for variables that were statistically significant in univariate analysis. The results of the 95% CI of HR were also presented visually as forest plots. The life table method was used to estimate the cumulative incidence of CRC per year of follow-up among subjects who had used metformin and statins prior to enrollment. All p values < 0.05 were considered statistically significant. The SAS statistical package (version 9.4; SAS Institute, Inc., Cary, NC, USA) and STATA (version 16.0; StataCorp, College Station, TX, USA) were used for all data analyses.

 Point 2.Due to the restrictions authors defined T2DM patients, the number of T2DM patients included in the study is very limited. In most areas of the world, between the age of 40-64, the percentage of T2DM patients is higher than the ~2% of the current study. Definition of T2DM must be redefined.

Response 2: In Page 3, Line 93-98, we modified the description as:

“The diabetic group was defined by records of at least three outpatient or inpatient diabetes diagnosis and prescribed antidiabetic medications within one year. The definition was according to the previous study from the Taiwan NHI Research Database[25], and the first diagnosis date was regarded as the index date. The other patients were classified into the non-diabetic group, and the date of their first visit to the doctor was regarded as the index date.”

In our study, the definition of diabetic group was mainly based on other published studies that also used Nationwide Health Insurance database as data source. We have cited the associated study as Reference No.25. According to Reference No.25: “We defined patients as having type 2 diabetes if they had at least three outpatient claims or one inpatient claim within 1 year. Using ICD-9 codes and the algorithm of at least three outpatient or one inpatient claims to define type 2 diabetes has been validated by previous study done in Taiwan”.

Besides, according to“Sheen YJ, Hsu CC et al,. Trends in prevalence and incidence of diabetes mellitus from 2005 to 2014 in Taiwan. J Formos Med Assoc. 2019 Nov;118 Suppl 2:S66-S73”. The incidence of diabetes mellitus in Taiwan from 2005 to 2014 was as follow:

2005

2006

2007

2008

2009

2010

2011

2012

2013

2014

P value

Incidence(%), 20-79 y/o

0.786

0.781

0.811

0.823

0.871

0.861

0.868

0.904

0.919

0.882

<0.001

In our study, 2% (87229/4018833) is the incidence of Type II DM (newly diagnosis Type II DM subjects between 2007-2009, total 3 years). It is compatible to Sheen YJ ‘s study. (The annual incidence is around 0.781-0.919% from 2005 to 2014.)

Point3.What was the rationale behind using propensity score matching? If authors want to conduct a nationwide analysis, the statistical analysis must be done on the whole population, not in a specific subset.

Response3: Dear reviewer, thanks for your comment and advice.

The reason we adopted propensity score matching is to reduce or eliminate the effects of confounding when using observational data. The purpose is to balance the persons with and without type 2 diabetes in each respective cohort and to augment their comparability. Because baseline characteristics often differ systematically between two groups.

We used whole national population for matching directly, not specific subset.

In Page 4, Line 106-109, we have added the propensity score matching part into Materials and Methods:

“To reduce the effects of potential confounders and sampling bias in the diabetic and non- diabetic groups, this research used a 1:4 ratio in propensity score matching(PSM) . It is a common technique for selecting controls with identical background covariates to minimize differences among groups of study participants.”

Point 4.Due to the restrictions used to define T2DM patients, the ones, who did not meet these definitions can cause bias if they were included in the non-T2DM population. As authors used propensity score matching, and these "non-T2DM" patients are probably much better matches compared to those of true non-T2DM ones, the results may have been introduced with a significant bias.

Response 4: Dear reviewer, thanks for your comment and advice.

As the explanationin question 1, the definition of T2DM in our study was according to the previous study from the Taiwan NHI Research Database. The incidence of T2DM in our study is compatible to the incidence of T2DM in study “Trends in prevalence and incidence of diabetes mellitus from 2005 to 2014 in Taiwan.”

It is a more strictly way to define diabetes group. In this situation,the people selected into the diabetes group actually have diabetes. Basically, it is impossible for people who don't have diabetes are misclassified in this group.

The relative risk of colorectal cancer for non-diabetes groups to diabetes group may be underestimated. However, it is a non-differential misclassification .

Point 5. Baseline description of the study population is missing. Also, the number of CRCs develeoped in the observation period is missing as well. From Table 2, one could assume this number is 2857, however, considering the long follow-up time and large baseline population, this number seem to bee too small (similar to that of the number of T2DM patients).

Response 5:Dear reviewer, thanks for your comment and advice. We have added the description in Page 5, Line 148-150:

A total of 396,521 women were included in this study. The number of CRCs developed in the observation period were 665 in the diabetes group and 2192 in the non-diabetes group (Figure 1).”

Besides, according to data of Taiwan Cancer Registry Report in 2019 (The 108th year of the "Republic Era) : https://www.hpa.gov.tw/Pages/Detail.aspx?nodeid=269&pid=14913

The incidence of CRC for 40-64 years old women is around : 50-100/100000

In our study, average incidence of CRC in our study was 2857/396521=0.0007=0.07%=70/100000, which is general compatible to the report.

 Point6. In Discussion, authors wrote that metformin usage was compared to non-diabetcs. Please explain, why? Metformin usage should be compared within the the diabetic subpopulation only!

Response 6: Dear reviewer, thanks for your friendly reminder. We are not careful enough. We have changed “non-diabetes group” to “non-users”.In Page 9, Line 228-230 :

“The effect of metformin on CRC is inconsistent between different studies. Knapen et al. revealed that metformin users had a 1.2-fold increased risk of colorectal cancer compared with a non-users.”

Point 7. Authors should consider dividing NSAID to ASA (aspirin) and other NSAIDs.

Response7: Dear reviewer, thanks for your friendly reminder. We have changed “NSAID” to “other NSAIDs” in this study.

 Point8. Based on T2DM medication, the advancement of T2DM could be predicted. It would be interesting to compare the different groups of T2DM patients in relation to CRC risk.

Response8: Dear reviewer, thanks for your comment and advice.

The current database has limitation to survey the severity or control situation of DM , we have added this part to our limitation in Page 10, Line 270-273.

Point 9. If authors can obtain the diagnosis date for T2DM from the database, the relationship between CRC and T2DM duration would be an interesting question.

Response9: Dear reviewer, thanks for your comment and advice. Although it is not the main issue we focused in this research, it is an interesting topic. Thanks for your guidance and inspiration.

Point 10. Figure 2: If the curves "touch" or cross each other, that is not cumulative incidence!

Response10: Dear reviewer, thanks for your comment and advice.

These are four different groups of people classified according to situation of metformin or statin use. The four curves represented the cumulative incidence of four different scenarios. During the observation period, we gradually estimate the annual incidence of CRC, which is cumulative incidence.  

Minor issues:

Point 1.There is a newer GLOBOCAN report.

Response1: Dear reviewer, thanks for your friendly reminder. In Page 1, Line 36-37, according to the GLOBOCAN 2020,we have modified the description to:

“Colorectal cancer (CRC) is the third most common malignancy and the second most common cause of cancer death worldwide.”

Point 2. There are some abbreviatoins throughout the manuscript, which were never solved within the main text (e.g., aHR and PAD).

Response2: Dear reviewer, thanks for your friendly reminder. We have checked the abbreviations problem.

Point 3. Authors wrote that the result on metformin vs CRC is controversial. This should be clarified as it is not limited to CRC only, but to cancer in general.

Response3: Dear reviewer, thanks for your comment and advice.

Some observational and cohort studies suggest metformin is associated with a reduced risk of a number of cancers, although the data is not conclusive. For example, study of Hyun Kyu Ahn et al.: “ Current Status and Application of Metformin for Prostate Cancer: A Comprehensive Review.” showed that some research has suggested that there is no association between metformin use and prostate cancer incidence or survival. The same situation is found in breast cancer. (Mónica Cejuela et al. Metformin and Breast Cancer: Where Are We Now?Int J Mol Sci. 2022 Feb 28;23(5):2705. ). The controversial results may due to different pathological mechanism, risk factor adjustment or study design.

Point4. In discussion it is not recommended to repeat numeric results and p-values.

Response 4:Dear reviewer, thanks for your friendly reminder. We have deleted the repeat numeric results and p-values of our study in discussion .

Point 5. An English proof-reading of a native speaker would be beneficial.

Response5: Dear reviewer, thanks for your friendly reminder. The revised manuscript has undergone English language editing by MDPI. The text has been checked for correct use of grammar and common technical terms, and edited to a level suitable for reporting research in a scholarly journal.

Reviewer 2 Report

The authors performed epidemiological studies on the relationship between Type II diabetes mellitus (T2DM) and female colorectal cancer (CRC), and metformin or statin effects on female CRC. In research conducted in the laboratory and on animals, metformin seems to protect against colorectal cancer. Metformin has also been demonstrated in epidemiological studies to have a preventive effect against colorectal cancer, which is unrelated to its anti-diabetic characteristics (CRC). However, there is still controversy about the effect of metformin and statins on CRC. This is an interesting research project. Unfortunately, I believe that the current research paper style is poor and not suitable for publication in MDPI's Clinical Medicine journal. First, however, here are my general and specific comments to the authors:

General comments: The author studied the effects of metformin and statins on female CRC in a population-based cohort study of women in Taiwan. Though it is informative and interesting research, but according to my opinion, the current study work style needs to improve for publishing in MDPI's Clinical Medicine journal. As a result, I believe the work might be accepted for publication in this Journal with just Major revisions.

Specific comments:

1. Abstract: "The incidence of colorectal cancer (CRC) in women has gradually increased in recent years." This is a confusing sentence. Please fix it.

2. Introduction: Please follow the referencing style according to the journal style. Please merge the last paragraph with the last sentences and rewrite the paragraph.

3. The author didn't follow the journal's instructions on how to put the article together.

4. In Figure 2, the data and results are confusing. Please double-check it.

5. In line 180, the author stated that NSAID use showed a protective effect on CRC in our study due to its anti-inflammatory effect, which is controversial in line 207 of the conclusion.

6. The materials and methods section is not clear.

7. Discussion: This part should be rewritten.

8. The conclusion part should be rewritten.

9. The English language and style must be extensively edited.

Author Response

Point 1. Abstract: "The incidence of colorectal cancer (CRC) in women has gradually increased in recent years." This is a confusing sentence. Please fix it.

Response 1: Dear reviewer, thanks for your comment and advice. The revised manuscript has undergone English language editing by MDPI. The text has been checked for correct use of grammar and common technical terms, and edited to a level suitable for reporting research in a scholarly journal.

In Page 1, Line 18-19. We have modified the sentence to: “In the last few years, the incidence of colorectal cancer (CRC) in women has gradually increased.”

Point 2. Introduction: Please follow the referencing style according to the journal style. Please merge the last paragraph with the last sentences and rewrite the paragraph.

Response 2: Dear reviewer, thanks for your friendly reminder. In Page 2, Line 68-71. We have modified the sentence to “ To our knowledge, there are controversial results and a lack of conclusive evidence in previous literature about the influence of metformin and statin on CRC. Therefore, this study aimed to investigate the effect of metformin and statin use on female CRC. “

Point 3. The author didn't follow the journal's instructions on how to put the article together.

Response 3: Dear reviewer, thanks for your comment and advice. We have modified this part.

Point 4. In Figure 2, the data and results are confusing. Please double-check it.

Response 4: Dear reviewer, thanks for your friendly reminder.

Figure 2 showed the cumulative incidence of female colorectal cancer based on metformin and statin use.

These are four different groups of people classified according to situation of metformin or statin use. The four curves represented the cumulative incidence of four different scenarios. During the observation period, we gradually estimate the annual incidence of CRC, which is cumulative incidence.  

“Number at risk” means number of individuals still under observation at the start of interval. The number may decrease year by year because of occurrence of CRC or individuals death from any cause, and the follow-up was stopped.

The cumulative incidence was described in Page 7, Line 174-181. 

Point 5. In line 180, the author stated that NSAID use showed a protective effect on CRC in our study due to its anti-inflammatory effect, which is controversial in line 207 of the conclusion.

Response 5: Dear reviewer, thanks for your friendly reminder. In Page 9, Line 253-254.

We have modified the description: “ NSAIDs use showed a protective effect on CRC in our study due to its anti-inflammatory effect. ”

Point 6. The materials and methods section is not clear.

Response 6: Dear reviewer, thanks for your comment and advice. Based on the reviewer's comprehensive suggestion, in Page 2-4, Line 73-145, we revised the methodology section and made more detailed additions:

  1. Materials and Methods

2.1. Data Source and Study Population

This was a propensity score-matched, retrospective cohort study based on a population of. women aged 40–64 years in Taiwan. The National Health Insurance (NHI) data. base, the Taiwan Cancer Registry (TCR), and the National Cause of Death Registry (NCDR) database, which are all available from the Health and Welfare Data Science Center (HWDC), were used in this study. The NHI database is an electronic, de-identified, administrative healthcare database containing the medical records of all outpatients and inpatients[23]. The national population-based TCR incorporates the diagnosis date of pathologically confirmed cancer cases into the registry and has high quality data [24]. Following the center's ethical guidelines, patients’ personal information was anonymized before we accessed it; consequently, the Research Ethics Committee waived the requirement for informed consent. This study was approved by the Institutional Review Board (IRB) and Ethics Committee of Fu Jen Catholic University (IRB approval no.: C107099).                       

2.2. Participant Selection

Figure 1 shows a flowchart of the selection procedure used in this study. A total of 4,018,833 Taiwanese women aged 40 to 64 (as CRC is rare in young people) who sought medical treatment from 2007 to 2009 were initially enrolled in the cohort, and was divided into diabetic and non-diabetic groups based on disease code and medication prescription. The diabetic group was defined by records of at least three outpatient or inpatient diabetes diagnosis and prescribed antidiabetic medications within one year. The definition was according to the previous study from the Taiwan NHI Research Database[25], and the first diagnosis date was regarded as the index date. The other patients were classified into the non-diabetic group, and the date of their first visit to the doctor was regarded as the index date. Patients who had had type 1 DM and CRC prior to the index date were excluded from our study. Definitions related to disease diagnosis and drugs are listed in Appendix Documents Tables A–C based on a previously published study in Taiwan[26].

2.3. Propensity Score Matching and Covariates

To reduce the effects of potential confounders and sampling bias in the diabetic and non- diabetic groups, this research used a 1:4 ratio in propensity score matching(PSM) . It is a common technique for selecting controls with identical background covariates to minimize differences among groups of study participants. The matching variables used were age, comorbidities (hypertension, chronic obstructive pulmonary disease (COPD), asthma, stroke, nephropathy, ischemic heart disease (IHD), peripheral arterial disease (PAD), eye disease, dyslipidemia, and obesity), concomitant medications (fibrates, angiotensin-converting enzyme inhibitor (ACEI), angiotensin receptor blocker (ARB), calcium channel blockers (CCB), aspirin, dipyridamole, clopidogrel, ticlopidine, and other non-steroidal anti-inflammatory drugs (NSAID)), and potential colon cancer detection examinations (definitions are listed in Appendix Documents Tables A–C and see Table 1). After PSM, the final cohort of patients with T2DM comprised 86,992 patients, whereas the comparison cohort contained 309,529 patients.

2.4. Statistical analyses

Basic demographic characteristics were defined as categorical (presented as N (%)) and con. tinuous variables (presented as mean ± standard deviation (SD)) in this study. The primary outcome in this study was the end point of CRC (ICD, 9th Revision, code 153). The person-years of follow-up (censoring time) for each participant were calculated from the index date until the occurrence of CRC, the date for all death, or the last date of linked data available from the TCR and NCDR (December 31, 2015), whichever occurred first.

We used a Cox proportional hazards regression to calculate hazard ratios (HRs) with 95% confidence intervals (CIs) between each risk factor and CRC. Variables were considered for more than three records in one year during the five years before the index date, including hypertension, chronic obstructive pulmonary disease (COPD), asthma, stroke, nephropathy, ischemic heart disease (IHD), peripheral arterial disease (PAD), eye disease, dyslipidemia, obesity, statin, fibrates, angiotensin-converting enzyme inhibitor (ACEI), angiotensin receptor blocker (ARB), calcium channel blockers (CCB), aspirin, dipyridamole, clopidogrel, ticlopidine, non-steroidal anti-inflammatory drugs (NSAID), sulfonylurea, insulin, acarbose, thiazolidinedione (TZD), potential colon cancer detection examinations, and metformin use. (see Appendix Documents Tables A–C).

Even if PSM is applied, residual imbalance might still exist in a population [27], and a multivariate Cox regression analysis should still be performed. Therefore, we performed a multivariate analysis for variables that were statistically significant in univariate analysis. The results of the 95% CI of HR were also presented visually as forest plots. The life table method was used to estimate the cumulative incidence of CRC per year of follow-up among subjects who had used metformin and statins prior to enrollment. All p values < 0.05 were considered statistically significant. The SAS statistical package (version 9.4; SAS Institute, Inc., Cary, NC, USA) and STATA (version 16.0; StataCorp, College Station, TX, USA) were used for all data analyses.analysis for variables that were statistically significant in univariate analysis. The results of the 95% CI of HR were also presented visually as forest plots. The life table method was used to estimate the cumulative incidence of CRC per year of follow-up among subjects who had used metformin and statins prior to enrollment. All p values < 0.05 were considered statistically significant. The SAS statistical package (version 9.4; SAS Institute, Inc., Cary, NC, USA) and STATA (version 16.0; StataCorp, College Station, TX, USA) were used for all data analyses.

Point 7. Discussion: This part should be rewritten.

Response 7: Dear reviewer, thanks for your friendly reminder. The revised manuscript has undergone English language editing by MDPI. The text has been checked for correct use of grammar and common technical terms, and edited to a level suitable for reporting research in a scholarly journal.

Point 8. The conclusion part should be rewritten.

Response 8: Dear reviewer, thanks for your friendly reminder. The revised manuscript has undergone English language editing by MDPI. The text has been checked for correct use of grammar and common technical terms, and edited to a level suitable for reporting research in a scholarly journal.

Point 9. The English language and style must be extensively edited.

Response 9: Dear reviewer, thanks for your friendly reminder. The revised manuscript has undergone English language editing by MDPI. The text has been checked for correct use of grammar and common technical terms, and edited to a level suitable for reporting research in a scholarly journal.

Round 2

Reviewer 1 Report

Although the revised manuscript significantly improved, the main issue of the article still persists. Authors should have excluded all patients from the non-diabetes cohort who even has the slightest suspicion of any disease related to the glucose homeostasis. Due to its operation, propensity score matching will favor those individuals who are more similar to each other. Consequently, it is inevitable that the algorythm will favor those "non-diabetic" patients with higher probability who are basically early diabetics but just require e.g. only dietary changes as treatment within the 3 year inclusion period. Therfore, authors should exclude at least those patients from the analysis, who developed diabetes per their definition during the follow-up period, but the exclusion of patients with all diabetes-related ICD codes would be more preferable.

A further question: How did authors handle the introduction or withdrawal of metformin/statin use during the observational period? A Cox regression with time dependent covariate can handle this situation perfectly.

In the caption of Figure 2, authors should indicate that they have overlayed the cummulative incidence curves of the 4 different subgroups on a single figure, and not the whole population CI was presented.

Author Response

Response to Reviewer 1 Comments

Point 1. Although the revised manuscript significantly improved, the main issue of the article still persists. Authors should have excluded all patients from the non-diabetes cohort who even has the slightest suspicion of any disease related to the glucose homeostasis. Due to its operation, propensity score matching will favor those individuals who are more similar to each other. Consequently, it is inevitable that the algorythm will favor those "non-diabetic" patients with higher probability who are basically early diabetics but just require e.g. only dietary changes as treatment within the 3 year inclusion period. Therfore, authors should exclude at least those patients from the analysis, who developed diabetes per their definition during the follow-up period, but the exclusion of patients with all diabetes-related ICD codes would be more preferable.

Response 1: Dear reviewer, thanks for your comment and advice.

It is a retrospective study. There is a possibility that the samples in the non-diabetes group develop diabetes later during the following period. However, the relative risk of colorectal cancer for non-diabetes groups to diabetes group was underestimated and more conservative. It is a non-differential misclassification .

Point 2. A further question: How did authors handle the introduction or withdrawal of metformin/statin use during the observational period? A Cox regression with time dependent covariate can handle this situation perfectly.

Response 2: Dear reviewer, thanks for your comment and advice.

In Page 4, Line 124-126, the censoring time for incidence of colorectal cancer is as follow:  “The person-years of follow-up (censoring time) for each participant were calculated from the index date until the occurrence of CRC, the date for all death, or the last date of linked data available from the TCR and NCDR (December 31, 2015), whichever occurred first.” Colorectal cancer (outcome variable, Y ) is a time dependent variable.

For independent variables (X), due to the limitation of the data, we use baseline data for risk estimation and time dependent estimation was not done. Thanks for reviewer’s advice and guidance.

Point 3. In the caption of Figure 2, authors should indicate that they have overlayed the cummulative incidence curves of the 4 different subgroups on a single figure, and not the whole population CI was presented.

Response 3: Dear reviewer, thanks for your comment and advice.

To avoid confusion, in Page 7, Line 184-185, we have changed the caption to “ Figure 2. Cumulative incidence of female colorectal cancer based on different scenarios of metformin and statin use.” From Figure 2, it can be observed that the cumulative incidence of these four groups does not overlap, showing a hazard constant that conforms to the cox proportional hazard model.
